# LARGE SCALE DISTRIBUTED NEURAL NETWORK TRAINING THROUGH ONLINE DISTILLATION

**Rohan Anil**
Google
rohananil@google.com

**Gabriel Pereyra** *
Google DeepMind
pereyra@google.com

**Alexandre Passos**
Google Brain
apassos@google.com

**Robert Ormandi**
Google
ormandi@google.com

**George E. Dahl**
Google Brain
gdahl@google.com

**Geoffrey E. Hinton**
Google Brain
geoffhinton@google.com

## ABSTRACT

Techniques such as ensembling and distillation promise model quality improvements when paired with almost any base model. However, due to increased test-time cost (for ensembles) and increased complexity of the training pipeline (for distillation), these techniques are challenging to use in industrial settings. In this paper we explore a variant of distillation which is relatively straightforward to use as it does not require a complicated multi-stage setup or many new hyperparameters. Our first claim is that online distillation enables us to use extra parallelism to fit very large datasets about twice as fast. Crucially, we can still speed up training even after we have already reached the point at which additional parallelism provides no benefit for synchronous or asynchronous stochastic gradient descent. Two neural networks trained on disjoint subsets of the data can share knowledge by encouraging each model to agree with the predictions the other model would have made. These predictions can come from a stale version of the other model so they can be safely computed using weights that only rarely get transmitted. Our second claim is that online distillation is a cost-effective way to make the exact predictions of a model dramatically more reproducible. We support our claims using experiments on the Criteo Display Ad Challenge dataset, ImageNet, and the largest to-date dataset used for neural language modeling, containing $6 \times 10^{11}$ tokens and based on the Common Crawl repository of web data.

## 1 INTRODUCTION

For large-scale, commercially valuable neural net training problems, practitioners would be willing to devote many more machines to training if it sped up training time dramatically or improved the quality of the final model. Currently, distributed stochastic gradient descent (SGD), in both its synchronous and asynchronous forms (Chen et al., 2016), is the dominant algorithm for large-scale neural network training across multiple interconnected machines. Unfortunately, as the number of machines increases, there are diminishing improvements to the time needed to train a high quality model, to a point where adding workers does not further improve training time. A combination of infrastructure limitations and optimization barriers constrain the scalability of distributed minibatch SGD. The overhead of communicating weight updates and the long tail of the machine and network latency distributions slow down execution and produce thorny engineering challenges. For the synchronous algorithm, there are rapidly diminishing returns from increasing the effective batch size (LeCun et al., 2012; Keskar et al., 2017). For the asynchronous algorithm, gradient interference from inconsistent weights can cause updates to thrash and even, in some cases, result in worse final accuracy or completely stall learning progress. The precise scalability limit for distributed SGD will depend on implementation details of the algorithm, specifics of the infrastructure, and the capabilities of the hardware, but in our experience it can be very difficult to scale effectively much beyond

---

*Work completed while G. Pereyra was a Google Brain resident.

a hundred GPU workers in realistic setups. No algorithm for training neural nets will be infinitely scalable, but even scaling a bit beyond the limits of distributed SGD would be extremely valuable.

Once we have reached the limits of adding workers to distributed SGD, we could instead use extra machines to train another copy of the model and create an ensemble to improve accuracy (or trade this accuracy for training time by training the members of the ensemble for fewer steps). As an added benefit, the ensemble will make more stable and reproducible predictions, which can be useful in practical applications. However, ensembling increases the cost at test time, potentially violating latency or other cost constraints. Alternatively, to get nearly the same benefits of the ensemble without increasing test time costs, we can distill (Hinton et al., 2015; Bucila et al., 2006) an $n$-way ensemble of models into a single still-servable model using a two-phase process: first we use $nM$ machines to train an n-way ensemble with distributed SGD and then use $M$ machines to train the servable student network to mimic the $n$-way ensemble. By adding another phase to the training process and using more machines, distillation in general increases training time and complexity in return for a quality improvement close to the larger teacher ensemble model.

We believe that the additional training costs, in terms of both time and pipeline complexity, discourage practitioners from using ensemble distillation, even though it almost always would improve results. In this work, we describe a simpler online variant of distillation we call codistillation. Codistillation trains $n$ copies of a model in parallel by adding a term to the loss function of the $i$th model to match the average prediction of the other models.

Through large-scale experiments we show that, compared to distributed SGD, codistillation improves accuracy and speeds up training by allowing the productive use of more computational resources even beyond the point where adding more workers provides no additional speedup for SGD. Specifically, codistillation provides the benefits of distilling an ensemble of models without increasing training time. Codistillation is also quite simple to use in practice compared to a multi-phase distillation training procedure. Multi-phase distillation tends to encourage human intervention between the training phases to decide when to stop training the ensemble and start distilling it into a single model. We also show that codistillation does not lose the reproducibility benefits of ensembles of neural networks, reducing churn in the predictions of different retrains of the same model. Reducing prediction churn can be essential when testing and launching new versions of a model in a non-disruptive way in an existing service, although it is not as well-studied in the academic machine learning community.

Given the obvious relationship to distillation, very similar algorithms to codistillation have been independently described by multiple researchers. For example, Zhang et al. (2017) describes another simultaneous distillation algorithm but does not investigate the benefits in the distributed training case and only presents it as a potential quality improvement over regular distillation. We view the experimental validation of codistillation at scale as the primary contribution of our work. Another contribution of this work is our exploration of different design choices and implementation considerations for codistillation which we believe has produced recommendations of substantial practical utility.

In general, we believe the quality gains of codistillation over well-tuned offline distillation will be minor in practice and the more interesting research direction is exploring codistillation as a distributed training algorithm that uses an additional form of communication that is far more delay tolerant.

## 1.1 RELATED WORK

In addition to the closely related work in Hinton et al. (2015) and Zhang et al. (2017) mentioned above, there are many different tactics for scaling up neural network training. Early work in training large distributed neural networks focused on schemes for partitioning networks over multiple cores, often referred to as model parallelism (Dean et al., 2012). As memory has increased on graphic processing units (GPUs), the majority of distributed training has shifted towards data parallelism, where the model is replicated across multiple machines and data are distributed to the different replicas, with updates being merged by parameter servers or a single allreduce step as in Goyal et al. (2017). Even without a high quality allreduce primitive, variants of centralized synchronous SGD with backup workers can scale to a large number of machines (Chen et al., 2016).

Methods like ensembling and distillation are mostly orthogonal to lower level distributed training infrastructure. However, mixture of experts models have particularly natural model parallelism that can be integrated with data parallelism and a synchronous training scheme. Gross et al. (2017) and Shazeer et al. (2017) are notable examples of recent work in this area.

As researchers try to scale neural network training to ever larger datasets and models, the optimization algorithm itself can be altered. For synchronous SGD there are rapidly diminishing returns (LeCun et al., 2012; Keskar et al., 2017) as the number of workers, and thus the effective batch size, increases and we might hope that algorithms like KFAC (Ba et al., 2017) would make better use of large batches. Although a promising direction for research, in this work we focus on what should hopefully be an optimization algorithm agnostic way to improve scalability and reproducibility.

## 2 CODISTILLATION

Distillation is a meta-algorithm which allows any algorithm to incorporate some of the model quality benefits of ensembles. The idea of distillation is to first train a *teacher model*, which traditionally is an ensemble or another high-capacity model, and then, once this teacher model is trained, train a *student model* with an additional term in the loss function which encourages its predictions to be similar to the predictions of the teacher model.

There are many variants of distillation, for different types of teacher model, different types of loss function, and different choices for what dataset the student model trains on. For example, the student model could be trained on a large unlabeled dataset, on a held-out data set, or even on the original training set.

Perhaps surprisingly, distillation has benefits even if the teacher model and the student model are two instances of the same neural network (see section 3 for empirical evidence), as long as they are sufficiently different (say, by having different initializations and seeing the examples in a different order). Furthermore, the teacher model predictions are still beneficial to the student model even before convergence. Finally, the distinction between teacher and student is unnecessary and two or more models all distilling from each other can also be useful.

In this paper, we use *codistillation* to refer to distillation performed:

1. using the same architecture for all the models;
2. using the same dataset to train all the models; and
3. using the distillation loss during training before any model has fully converged.

For simplicity, we usually consider the case where all models have a distillation term in their loss function, but the key characteristic of codistillation is the simultaneous training of a model and its teacher.

Algorithm 1 presents the codistillation algorithm. The distillation loss term $\psi$ can be the squared error between the logits of the models, the KL divergence between the predictive distributions, or some other measure of agreement between the model predictions. In this work we use the cross entropy error treating the teacher predictive distribution as soft targets. In the beginning of training, the distillation term in the loss is not very useful or may even be counterproductive, so to maintain model diversity longer and to avoid a complicated loss function schedule we only enable the distillation term in the loss function once training has gotten off the ground.

### 2.1 CODISTILLATION AS A DISTRIBUTED NEURAL NETWORK TRAINING ALGORITHM

In order to scale beyond the limits of distributed stochastic gradient descent we will need an algorithm that is far more communication efficient. As seen in Algorithm 1, to update the parameters of one network using codistillation one only needs the predictions of the other networks, which can be computed locally from copies of the other networks weights.

There are several reasons to believe that stale predictions might be much less of a problem than stale gradients for training:

---

**Algorithm 1** Codistillation

---
**Input** loss function $\phi(\texttt{label}, \texttt{prediction})$
**Input** distillation loss function $\psi(\texttt{aggregated\_label}, \texttt{prediction})$
**Input** prediction function $F(\theta, \texttt{input})$
**Input** learning rate $\eta$
**for** $\texttt{n\_burn\_in}$ steps **do**
    **for** $\theta_i$ in $\texttt{model\_set}$ **do**
        $\texttt{y}, \texttt{f}$ = $\texttt{get\_train\_example()}$
        $\theta_i = \theta_i - \eta\nabla_{\theta_i}\{\phi(y, F(\theta_i, f))\}$
    **end for**
**end for**
**while** not converged **do**
    **for** $\theta_i$ in $\texttt{model\_set}$ **do**
        $\texttt{y}, \texttt{f}$ = $\texttt{get\_train\_example()}$
        $\theta_i = \theta_i - \eta\nabla_{\theta_i}\{\phi(\texttt{y}, F(\theta_i, f)) + \psi(\{\frac{1}{N-1}\sum_{j \neq i} F(\theta_j, f)\}, F(\theta_i, f))\}$
    **end for**
**end while**

---

1. every change in weights leads to a change in gradients, but as training progresses towards convergence, weight updates should substantially change only the predictions on a small subset of the training data;

2. weights (and gradients) are not statistically identifiable as different copies of the weights might have arbitrary scaling differences, permuted hidden units, or otherwise rotated or transformed hidden layer feature space so that averaging gradients does not make sense unless models are extremely similar;

3. sufficiently out-of-sync copies of the weights will have completely arbitrary differences that change the meaning of individual directions in feature space that are not distinguishable by measuring the loss on the training set;

4. in contrast, output units have a clear and consistent meaning enforced by the loss function and the training data.

Furthermore, the predictive distribution of radically different models can still provide very useful information about the relationship between inputs and outputs. Empirically we've found that using stale predictions instead of up-to-date predictions for the other neural networks has little to no adverse effect on the quality of the final trained model produced by codistillation. We have been able to use predictions tens of thousands of updates old in the asynchronous case or 800k examples (i.e. 50 updates) old in the large-batch synchronous case.

The tolerance of distillation for stale teacher predictions suggests a distributed training strategy which is far less communication-intensive than synchronous or asynchronous SGD.

1. Each worker trains an independent version of the model on a locally available subset of the training data.

2. Occasionally, workers checkpoint their parameters.

3. Once this happens, other workers can load the freshest available checkpoints into memory and perform codistillation.

Of course there is no reason not to combine this strategy with standard distributed SGD, resulting in a procedure that employs independent groups of workers that exchange checkpoints between groups and exchange gradient information within a group.

In each iteration of synchronous/asynchronous distributed SGD, each worker needs to send and receive an amount of information proportional to the entire model size. When using codistillation to distribute training each worker only needs to very rarely read parameter checkpoints from the other models.[1] When combining distributed SGD and codistillation, we can add workers to a group up

---

[1] Although our implementation of codistillation exchanges model checkpoints, there are some cases where an alternative communication approach would be desirable. One obvious alternative would be to use a predic-

until the point where we see diminishing returns from distributed SGD and then deploy additional workers in another group, occasionally exchanging checkpoints between the otherwise independent groups of workers.

Moreover, there is no need to use high-precision floating point numbers to store the parameters used to compute the predictions for the distillation loss term as distillation is not very sensitive to the exact values of the predictions. Therefore the additional computational cost of distributed codistillation will not be much higher than the cost of independent training.

Since the parameters of a model trained on a data set can be viewed as a very compressed representation of the aspects of that data set which are relevant to the learning problem at hand, it makes intuitive sense that leveraging these parameters might be more communication-efficient than sending all the data points or gradients.

## 3 EXPERIMENTS AND RESULTS

In order to study the scalability of distributed training using codistillation, we need a task that is representative of important large-scale neural network training problems. Neural language modeling is an ideal test bed because vast quantities of text are available on the web and because neural language models can be very expensive to train. Neural language models are representative of important problems that make common use of distributed SGD like machine translation and speech recognition, but language modeling is simpler to evaluate and uses a simpler pipeline. In order to make any potential scalability improvements as clear as possible, we selected a data set large enough that it is completely infeasible to train an expressive model to convergence on it with existing SGD parallelization strategies. In order to confirm that our results were not specific to some peculiarity of language modeling, we also validated some of our large scale codistillation results on ImageNet (Russakovsky et al., 2015) as well.

To demonstrate the benefits of codistillation in reducing prediction churn and to study other properties of the algorithm we can use smaller experiments that are cheaper to perform, but it is important to actually reach the limits of distributed SGD when studying scalability.

### 3.1 DATA SETS AND MODELS

**Common Crawl** is an open repository of web crawl data. We downloaded the WET files[2] and filtered them to only include English language documents that contained long paragraphs because we wanted data that allowed modeling of slightly longer range dependencies than data sets that randomize sentence order like LM1B (Chelba et al., 2013). After preprocessing, roughly 915 million documents (20TB of text) remained. We plan to release the list of document ids that remained after filtering as well as code for our invertible tokenizer for others to use this data set. The language model we trained in all of our Common Crawl experiments was an RNN language model with two LSTM layers of 1024 units each with layer normalization (Ba et al., 2016). We used 256 dimensional input embeddings and a vocabulary of 24006 word pieces (Schuster & Nakajima, 2012), including sentence and paragraph start and end tokens, out of vocabulary (generally non-English characters), and end of document. After converting to word pieces there were 673 billion tokens, which is much larger than any previous neural language modeling data set we are aware of.[3] During training we constructed batches 32 word pieces long drawing tokens from $B$ different documents at a time, saving hidden state across batches. Since the hidden state never gets reset, the model has to learn to use the end of document token to reset itself. We use the ADAM optimizer Kingma & Ba (2014) for all experiments on Common Crawl.

---

tion server to communicate predictions instead of weights. Workers could read teacher predictions along with a minibatch of data and send their predictions back to the server after each update or separate, evaluation-only workers could read checkpoints and continuously update the predictions for each piece of training data. This strategy might be most appropriate in the presence of specialized forward-pass hardware. Another alternative to communicating checkpoints is to train all copies of the model in the same process which would make the most sense when the size of the model relative to the characteristics of the hardware make it almost free to to run both models.

[2] http://commoncrawl.org/2017/07/june-2017-crawl-archive-now-available/

[3] It is still infeasible to train on all of this data set with large neural language models, so our experiments did not use all of it, but we hope it will be useful for future scalability research.

**ImageNet** is the most popular image classification benchmark of recent years. All of our experiments on ImageNet followed the setup of Goyal et al. (2017) as closely as possible and also used fully-synchronous SGD. We used the same learning rate scaling and schedule and used a configuration with batch size 16384 that achieves 75% accuracy as our primary baseline. Goyal et al. (2017) reports that increasing the batch size beyond 8192 provides rapidly diminishing returns.

**Criteo Display Ad Challenge** dataset[4] (Criteo) is a benchmark dataset for predicting click through rate for online advertising. The data contain roughly 43 million examples, each with 13 integer and 26 categorical input features. The task is formulated as binary classification and we train a feed-forward fully connected neural network using ReLU activations with hidden layer sizes of 2560, 1024, 256 and a logistic output. We use the Adagrad optimizer with learning rate of 0.001 for training for all experiments on this dataset.

### 3.2 Reaching the limits of distributed SGD for training RNNs on Common Crawl

In our first set of experiments, our goal was to approximately determine the maximum number of GPU workers that can be productively employed for SGD in our Common Crawl neural language model setup. Since our dataset is two orders of magnitude larger than English Wikipedia, there is no concern about revisiting data, which would make independent replicas more similar, even in relatively large scale experiments.

We tried asynchronous SGD with 32 and 128 workers, sharding the weights across increasing numbers of parameter servers as necessary to ensure that training speed was bottlenecked by GPU computation time. We found it very difficult to keep training stable and prevent the RNNs from diverging for asynchronous SGD with large numbers of workers. We experimented with a few worker ramp up schemes and different learning rates, but ultimately decided to focus on the synchronous algorithm to make our results less dependent on the specific characteristics of our infrastructure and implementation. Gradient staleness is hard to analyze independent of the specific conditions whereas differences in implementation and infrastructure are far easier to abstract away for synchronous SGD. Although it may have been possible to make async work well with more effort, the debilitating effect of stale gradients on learning progress is a well known issue, for instance Chen et al. (2016) demonstrated that synchronous SGD can often converge to a better final accuracy than asynchronous SGD. Mitliagkas et al. (2016) argues that asynchrony can effectively increase the momentum which is part of why it tends to diverge so easily. For these and other reasons, practitioners (e.g. Goyal et al. (2017)) seem to be moving away from asynchronous SGD towards synchronous training as the default. In preliminary experiments the gains from codistillation seemed independent of the choice of asynchronous or synchronous SGD as the base algorithm.

The maximum number of GPU workers that can be productively employed for synchronous SGD will depend on infrastructure limits, tail latency, and batch size effects. Fully synchronous SGD is equivalent to the single machine algorithm with a much larger batch size. Increasing the effective batch size reduces noise in the gradient estimates which allows larger step sizes with hopefully higher quality updates that result in faster convergence. Given effectively infinite training data (even with 256 GPUs we do not visit all of the Common Crawl training data) we intuitively would expect increasing the effective batch size to at worst increase the step time. We trained language models on Common Crawl with fully synchronous SGD using a per-worker batch size of 128 and 32, 64, 128, and 256 workers. Thus the effective batch size ranged from 4096 to 32768. Generally we should expect to need to increase the learning rate as we increase the effective batch size, so for each number of workers we tried learning rates of 0.1, 0.2, and 0.4. For 32 and 64 workers, 0.1 performed best and since none of the original three learning rates performed well for 256 workers, we also tried an additional intermediate learning rate of 0.3 which was the best performing learning rate for 256 workers.

Figure 1a plots the validation error as a function of global steps for the different numbers of workers we tried, using the best learning rate for each number of workers. Increasing the number of workers (and thus the effective batch size) reduced the number of steps required to reach the best validation error until 128 workers, at which point there was no additional improvement. Even with idealized perfect infrastructure, 256 workers would at best result in the same end to end training time on this

---

[4]https://www.kaggle.com/c/criteo-display-ad-challenge

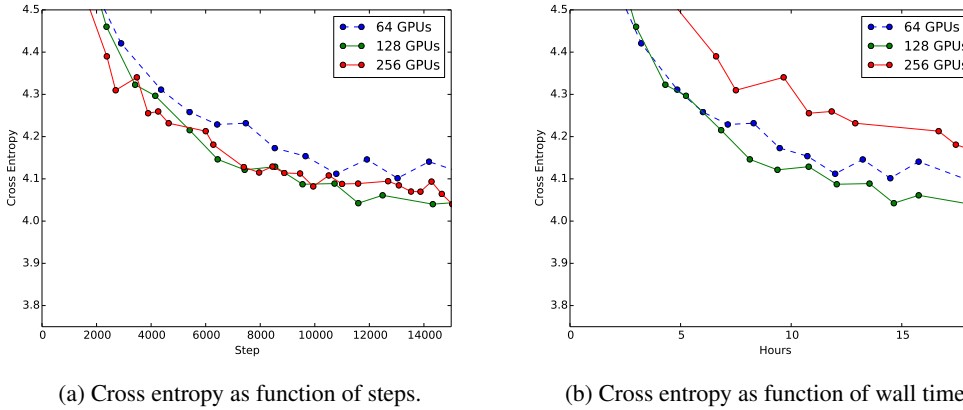

(a) Cross entropy as function of steps.                    (b) Cross entropy as function of wall time.

Figure 1: Synchronous training on Common Crawl dataset.

problem. However, because steps can take so much longer with 256 workers, going from 128 to 256 workers is highly counterproductive in practice. Figure 1b plots validation error against wall time for the same varying numbers of synchronous workers. There is a large degradation in step time, and thus learning progress, at 256 workers. Although it might be possible to improve the step time at 256 workers by using a more sophisticated scheme with backup workers (Chen et al., 2016), the operative limit to scalability on this task is the diminishing return from increasing the effective batch size, not the degradation in step times.

In these particular experiments, synchronous SGD with 128 workers is the strongest baseline in terms of training time and final accuracy. Therefore we focus the rest of our experiments on comparisons with 128 worker synchronous SGD and study codistillation that uses synchronous SGD as a subroutine, although it also works well with the asynchronous algorithm.

### 3.3 CODISTILLATION WITH SYNCHRONOUS SGD

For language modeling on Common Crawl, synchronous SGD with 128 GPUs achieved the best results for standard distributed training, at least of the configurations we tried, and we were unable to improve training time with 256 GPUs. Although the extra GPUs do not seem to help basic synchronous SGD, our hypothesis is that the extra 128 GPUs will improve training time if we use two-way codistillation with two groups of 128 GPUs using synchronous SGD that exchange checkpoints periodically.

One concern would be that codistillation is merely a way of penalizing confident output distributions (Pereyra et al., 2017) or smoothing the labels, so we also compared to two label smoothing baselines. The first baseline replaces the distillation loss term with a term that matches the uniform distribution and the second uses a term that matches the unigram distribution. Trade-off hyperparameters were tuned by hand in preliminary experiments.

Another important comparison is to an ensemble of two neural networks, each trained with 128 GPUs and synchronous SGD. Although we are in general interested in the regime where such an ensemble would not be practical because of the increased test time costs, given our understanding of distillation we would expect codistillation, if it achieves all of the benefits of traditional distillation, to have a training curve close to—but slightly worse than—a two-way ensemble. In the case of two-way codistillation, this would provide evidence that the gains are really coming from an ensembling-like effect despite never explicitly averaging the predictions of multiple models as would happen when distilling an ensemble model.

Figure 2a plots validation cross entropy versus step of synchronous training for codistillation using two groups of 128 GPUs along with training curves for the synchronous SGD and label smoothing baselines (each using 128 GPUs) and an ensemble of two instances of the 128 GPU baseline. All experiments in figure 2a used the learning rate found to be the best for the 128 GPU synchronous SGD baseline. Two-way codistillation successfully reduces training time substantially compared to

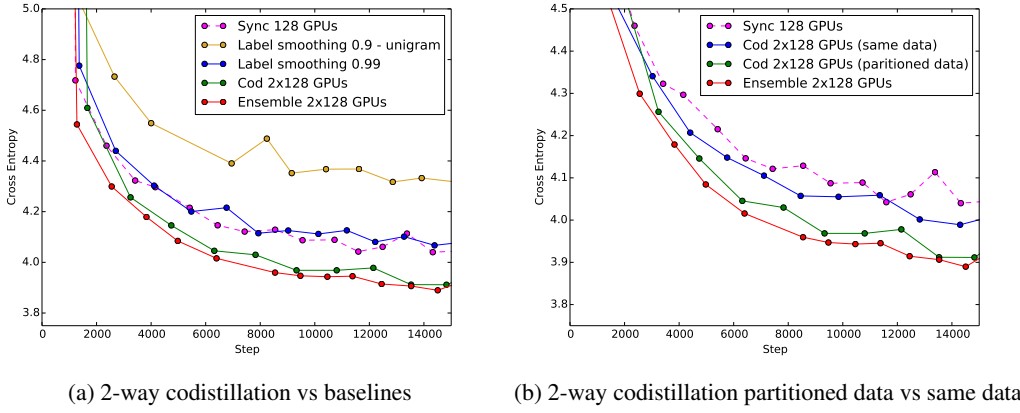

(a) 2-way codistillation vs baselines      (b) 2-way codistillation partitioned data vs same data

Figure 2: Codistillation (abbreviated as "Cod" in the legend) results with Common Crawl.

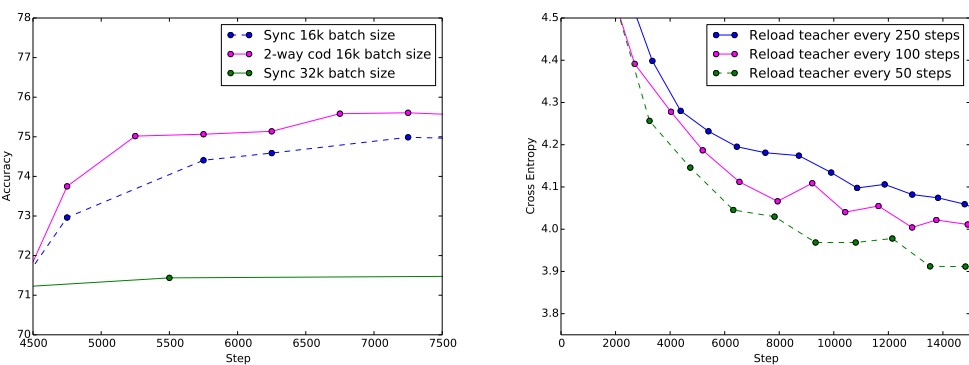

Figure 3: Codistillation on ImageNet      Figure 4: Reload intervals on Common Crawl

the 128 worker baselines and almost achieves the training curve of the two-way ensemble. Measuring at the best validation error achieved by the baseline, codistillation reaches the same error in 2X fewer steps. Perhaps more importantly, codistillation reaches a lower final error so a 2X reduction in steps is likely an underestimate of the gains. In our implementation, for the model we trained, codistillation is free in terms of step time as the GPU is not fully utilized and our implementation automatically overlaps the computation of the teacher and student models. In the worst case, for a model that saturates the hardware that is implemented without quantization, prefetching predictions using the CPU, or other optimizations to compute the predictions, the extra forward pass might increase compute costs by nearly 50%. However, even with these worst case assumptions, network costs will be a substantial contributor to the total step time, easily 50%-80%, resulting in a modest increase in time per step.

### 3.3.1 CODISTILLATION REQUIRES FEWER STEPS ON IMAGENET

In order to confirm our results are not due to the vicissitudes of language modeling or the particulars of our setup, we tried two-way codistillation on ImageNet as well. As can be seen from figure 3, codistillation (enabled after 3000 steps) achieves the same 75% accuracy after a total of 5250 steps as the baseline does after 7250 steps. Eventually, two-way codistillation achieves a slightly better accuracy of 75.6% at 7250 steps, confirming the utility of codistillation on ImageNet in addition to language modeling.

### 3.3.2 PARTITIONING THE DATASET

The gains from codistillation have a variety of potential causes that we would like to tease apart. From the experiments above, we have evidence against label smoothing effects as a cause. Another potential source of the training time and quality gains from codistillation over basic SGD would be that the different codistilling models see different training data. To test this hypothesis, we ran another codistillation experiment using two groups of 128 workers, but forced the two groups to use the same training data instead of using different random subsets.[5] Figure 2b compares codistillation using different subsets of the training data to codistillation using the same data. Codistillation with the same data seems to be slightly better than the baseline, but codistillation using different data gets much better results. These results show that the codistilling models are indeed successfully transmitting useful information about different parts of the training data to each other.

### 3.4 PREDICTION STALENESS SENSITIVITY

In general, codistillation can handle relatively stale predictions. SGD generally cannot tolerate gradients nearly as stale. The contrast is most stark with the asynchronous algorithm. We trained our two-way synchronous codistillation setup on Common Crawl with several different checkpoint exchange frequencies. We tried exchange delays of 50, 100, and 250 steps. As can be seen from figure 4, increasing the checkpoint reload interval beyond 819,200 examples or 50 steps slightly degrades the learning curve. With an interval of 50 steps, communicating checkpoints with a shared file system is still quite feasible on most problems.

#### 3.4.1 CODISTILLATION VS MULTI-PHASE DISTILLATION VARIANTS

In our experiments and experience, the choice between multi-phase distillation and codistillation makes very little difference in terms of the quality improvements achieved by distillation, although it does affect training time. On Common Crawl, we trained an ensemble of two models for 18K steps and then trained an identical model distilling from the ensemble to reach a cross entropy error of 4.0 after 9K steps for a total of 27K steps. However, two-way codistillation reached roughly the same validation error after only 10k steps. Zhang et al. (2017) reported a benefit in quality over basic distillation, but they compare distilling model $M_1$ into model $M_2$ with training model $M_1$ and model $M_2$ using codistillation; they do not compare to distilling an ensemble of models $M_1$ and $M_2$ into model $M_3$. Furthermore, Zhang et al. (2017) only reports final validation error using a large number of training steps on two small image datasets. Overfitting in the teacher model can explain the worse performance of offline distillation Zhang et al. (2017) report on CIFAR-100. We reproduced their experiments with WideResnet-28-10 teaching Resnet-32 on CIFAR-100. When we select a checkpoint with nearly 100% training accuracy, we reproduce the 69.5% they report in table 4. However, when we select a different checkpoint, we can achieve the 70.7% they report for online distillation using traditional offline distillation. For distillation to provide value, the teacher must provide information beyond the training label. Thus a teacher network that overfits the training set will not be useful.

The gains in training time of codistillation over multi-phase distillation variants are obvious, but the reduction in training pipeline complexity codistillation provides can be equally important. Using the same architecture and dataset for all the models avoids squaring the tuning problem. The codistillation protocol simplifies the choice of teacher model and restores symmetry between the various models. With traditional multi-phase distillation one must decide which teacher model or models to use and how long to train them, encouraging human intervention between the phases. If teacher models occasionally get reused across versions of the student model in an effort to save computation, rollbacks of models to deal with bugs or data corruption can be dramatically more painful. To reproduce a given final model, one needs the entire history of teacher models and everything required to reproduce them which can easily result in what Sculley et al. (2014) refers to as "pipeline jungle" and unnecessary, undesirable data dependencies.

---

[5]Even with codistillation and 256 GPUs, we still only visit 2.3% of the Common Crawl data. Our scalability experiments are motivated (and hopefully informative for) important practical problems like translation and Wu et al. (2016) also reports that they were unable to train on all the data they had available. Presumably, training larger models on more data would produce better results.

Table 1: Prediction Churn

| Model | Validation Log Loss | Mean Absolute Difference[6] Between Retrains |
|---|---|---|
| DNN | $0.4480 \pm 0.001$ | $0.029 \pm 0.001$ |
| Ensemble of Two DNNs | $0.4461 \pm 0.0002$ | $0.022 \pm 0.002$ |
| Two-way codistilled DNN | $0.4458 \pm 0.002$ | $0.019 \pm 0.002$ |

## 3.5 REDUCING PREDICTION CHURN WITH CODISTILLATION

Unlike linear models with convex loss functions, two neural networks with the same architecture that are trained on the same data can achieve similar validation and test performance while making very different predictions, and mistakes. On large datasets with a stable training pipeline aggregate metrics can be relatively consistent, but minor changes to the model architecture or even simple retrains can cause comparatively dramatic changes in the predictions made by the network. The network will in general get different examples correct and the differences might be especially severe on atypical examples with rare features. The weights learned by stochastic gradient descent in the non-convex setting will depend on the initialization, data presentation order, and the general vicissitudes of the infrastructure, especially when parallelization is involved. It is not practical to control all these nuisance variables and, even if it was, we would still see different solutions after making slight changes to the input representation or model architecture. We will refer to the general reproducibility problem where retraining a neural network after a minor (or even no) change causes a change to the predictions as *prediction churn*. Prediction churn can be a serious problem when testing and launching new versions of a neural network in a non-disruptive way inside an existing service.

Model averaging is a very natural solution to prediction churn. By directly averaging away the variations in the training procedure the predictions will tend to be more consistent across retrains of the ensemble and from minor modifications to the base models.

Given that codistillation achieves many of the benefits of model averaging, our hypothesis is that it should similarly help reduce prediction churn. To test this hypothesis, we trained a deep neural network (DNN) on the Criteo dataset and measured the mean absolute difference between the predictions of two retrains of the same model (prediction difference, for short). The prediction differences between different versions of a model should be at least as large as the prediction differences between two retrains of the same model and serve as a way of estimating the prediction churn. We also trained an ensemble of two copies of the initial DNN and then measured the prediction difference between retrains of the ensemble. Finally, we trained the same DNN using two-way codistillation, picking one of the copies arbitrarily to make predictions and measured the reproducibility of codistillation as well. As shown in table 1, codistillation reduces prediction churn by 35%, achieving similar results to ensembling, but does not increase serving costs.

## 4 DISCUSSION AND FUTURE WORK

Distillation is a surprisingly flexible tool, especially when performed during model training instead of after. It can be used to accelerate training, improve quality, distribute training in new, more communication efficient ways, and reduce prediction churn. However, there are still many questions we would like to explore. For example, we mostly focused on pairs of models codistilling from each other. It stands to reason that if pairs are useful then so are other topologies. Fully connected graphs might make the models too similar, too quickly so ring structures might also be interesting. We also did not explore the limits of how accurate the predictions from the teacher models have to be. It might be possible to aggressively quantize the teacher model to make codistillation almost as cheap as normal training even for very large models.

---

[6]In all cases we repeat the experiment five times and report the mean $\pm$ half the range.

It is somewhat paradoxical that bad models codistilling from each other can learn faster than models training independently. Somehow the mistakes made by the teacher model carry enough information to help the student model do better than the teacher, and better than just seeing the actual label in the data. Characterizing the ideal properties of a teacher model is an exciting avenue for future work.

In this work we only extract predictions from the checkpoints, as predictions are identifiable and, unlike the internal structure of the networks, have no spurious symmetries. That said, it might be possible to extract more information from a checkpoint than just predictions without hitting the same issues faced by workers communicating gradients, allowing the use of the teacher models as a stronger regularizer. Perhaps distillation-based methods could be used to augment federated learning McMahan et al. (2017) in particularly bandwidth-constrained settings.

ACKNOWLEDGMENTS

We would like to thank Avital Oliver for feedback on a draft and Oriol Vinyals for many helpful discussions. We would also like to thank Mohammad Norouzi for emotional support and Dan Hurt for essential help resolving last-minute computational resources issues.

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
