# OpenReview forum: "Large scale distributed neural network training through online distillation"
_ICLR.cc/2018/Conference — Accept (Poster)_

### Official Review · AnonReviewer3 · 2017-11-25
**Promising and interesting direction to scale distributed training**

**Rating:** 8
**Confidence:** 4

**Review:**

This paper provides a very original & promising method to scale distributed training beyond the current limits of mini-batch stochastic gradient descent. As authors point out, scaling distributed stochastic gradient descent to more workers typically requires larger batch sizes in order to fully utilize computational resource, and increasing the batch size has a diminishing return. This is clearly a very important problem, as it is a major blocker for current machine learning models to scale beyond the size of models and datasets we currently use. Authors propose to use distillation as a mechanism of communication between workers, which is attractive because prediction scores are more compact than model parameters, model-agnostic, and can be considered to be more robust to out-of-sync differences. This is a simple and sensible idea, and empirical experiments convincingly demonstrate the advantage of the method in large scale distributed training.

I would encourage authors to experiment in broader settings, in order to demonstrate that the general applicability of the proposed method, and also to help readers better understand its limitations. Authors only provide a single positive data point; that co-distillation was useful in scaling up from 128 GPUs to 258 GPUs, for the particular language modeling problem (commoncrawl) which others have not previously studied. In order for other researchers who work on different problems and different system infrastructure to judge whether this method will be useful for them, however, they need to understand better when codistillation succeeds and when it fails. It will be more useful to provide experiments with smaller and (if possible) larger number of GPUs (16, 32, 64, and 512?, 1024?), so that we can more clearly understand how useful this method is under the regime mini-batch stochastic gradient continues to scale. Also, more diversity of models would also help understanding robustness of this method to the model. Why not consider ImageNet? Goyal et al reports that it took an hour for them to train ResNet on ImageNet with 256 GPUs, and authors may demonstrate it can be trained faster.

Furthermore, authors briefly mention that staleness of parameters up to tens of thousands of updates did not have any adverse effect, but it would good to know how the learning curve behaves as a function of this delay. Knowing how much delay we can tolerate will motivate us to design different methods of communication between teacher and student models.

---

> ### Author Response · Authors · 2017-12-06
> **Thank you for your review**
>
> Thank you for your review. We have demonstrated large improvements on many non-public datasets that we were unable to include, so your point is well taken. We will try to add some ImageNet results. If we can get them done in time, we will update the paper and this discussion. We agree that these would greatly strengthen the paper.
>
> Unfortunately, we will not be able to move to a larger number of GPUs during the review period. However, given the limits of the baselines, it would probably not be much better to use more GPUs than we did in the CommonCrawl experiments.
>
> We will run additional experiments with different checkpoint exchange intervals to investigate the question about delay sensitivity in more depth.

---

> ### Author Response · Authors · 2017-12-23
> **updated experiments**
>
> We were able to add the ImageNet experiments you suggested, as well as experiments on the staleness of predictions.
>
> ImageNet results can be found in sections 3.1, 3.3.1, and 3.4.1 as well as figure 3 of the latest version.
>
> Section 3.4 and figure 4 now cover the prediction staleness issue.

---

### Official Review · AnonReviewer1 · 2017-11-27
**Incremental algorithmic novelty and limited experiments**

**Rating:** 4
**Confidence:** 3

**Review:**

The paper proposes an online distillation method, called co-distillation, where the two different models are trained to match the predictions of other model in addition to minimizing its own loss. The proposed method is applied to two large-scale datasets and showed to perform better than other baselines such as label smoothing, and the standard ensemble.

The paper is clearly written and was easy to understand. My major concern is the significance and originality of the proposed method. As written by the authors, the main contribution of the paper is to apply the codistillation method, which is pretty similar to Zhang et. al (2017), at scale. But, because from Zhang's method, I don't see any significant difficulty in applying to large-scale problems, I'm not sure that this can be a significant contribution. Rather, I think, it would have been better for the authors to apply the proposed methods to a smaller scale problems as well in order to explore more various aspects of the proposed methods including the effects of number of different models. In this sense, it is also a limitation that the authors showing experiments where only two models are codistillated. Usually, ensemble becomes stronger as the number of model increases.

---

> ### Author Response · Authors · 2017-12-06
> **Thank you for your review**
>
> Thank you for your review. We agree that online distillation is quite straightforward to come up with, in fact Geoff Hinton described the idea in a 2014 talk (https://www.youtube.com/watch?v=EK61htlw8hY). However, he told us that he did not publish the idea in a paper because numerous subsequent experiments showed that it did not outperform distilling an ensemble into a new model.  Appreciating the real practical benefit of codistillation (extra parallelism without needing a subsequent distillation phase) over offline distillation and demonstrating it at scale is far from trivial because it is essential to first exhaust simpler forms of parallelism.  For instance, just from reading Zhang et al., no one would want to use online distillation because the authors do not claim any training time improvement or other benefit beyond a small and dubious quality improvement. Since the submission deadline, we have investigated and reproduced some of Zhang et al.'s experiments and we now believe that overfitting in the teacher model mostly explains the worse performance of regular, offline distillation that they report. As far as we know, Zhang et al. is an unreviewed manuscript draft that, unlike our work, does not provide clear evidence for the benefits of online distillation.
>
> Our contribution is to articulate and demonstrate the practical value of online distillation algorithms, including benefits to reproducibility and training speed. We demonstrate these benefits at scale, in a hopefully convincing fashion. If our paper did not exist, it might be many years before people tried these algorithms again, because reading Zhang et al. alone makes it seem like a modest quality improvement is the only benefit (a quality improvement that becomes even smaller when the teacher model in offline distillation does not overfit).

---

### Official Review · AnonReviewer2 · 2017-11-30
**Clear analysis, good motivation and sufficient verification**

**Rating:** 6
**Confidence:** 3

**Review:**

Although I am not an expert on this area, but this paper clearly explains their contribution and provides enough evidences to prove their results.
Online distillation technique is introduced to accelerate traditional algorithms for large-scale distributed NN training.
Could the authors add more results on the CNN ?

---

> ### Author Response · Authors · 2017-12-06
> **Thank you for your review**
>
> Thank you for your review. Were there any specific results you think would be particularly useful? We would like to add some results with CNNs on ImageNet, but these are somewhat expensive, so we will update the thread if we can get them done to our satisfaction.
>
> We have some CIFAR results with CNNs we plan to add to help explain what we said in the manuscript about Zhang et al.'s results.

---

> > ### Author Response · Authors · 2017-12-23
> > **experiments added**
> >
> > We have added the CNN results on ImageNet and (brief) results on CIFAR-100 to the paper.
> > They can now be found in sections 3.1, 3.3.1, and 3.4.1 as well as figure 3.

---

### Author Response · Authors · 2017-12-21
**Revised manuscript should address reviewer concerns**

In response to reviewer feedback, we have improved the manuscript and at this point we believe the new version addresses all reviewer concerns.

We have added results on ImageNet that show that codistillation works there as well, even though it is a very different problem from language modeling. We achieve a state of the art number of steps to reach 75% accuracy on ImageNet.

We also reran experiments from Zhang et al. on CIFAR-100 and show that online and offline distillation actually produce the same accuracy, when offline distillation is done correctly, contrary to what their table shows. These results support our claim that our paper is the first to show the true benefits of online distillation.

We have also added delay sensitivity experiments on Common Crawl.

---

### Decision · Program_Chairs · 2018-01-29
**ICLR 2018 Conference Acceptance Decision**

**Decision:**

Accept (Poster)

**Comment:**

meta score: 7

The paper introduces an online distillation technique to parallelise large scale training.  Although the basic idea is not novel, the presented experimentation indicates that the authors' have made the technique work.  Thus this paper should be of interest to practitioners.

Pros:
 - clearly written, the approach is well-explained
 - good experimentation on large-scale common crawl data with 128-256 GPUs
 - strong experimental results

Cons:
 - the idea itself is not novel
 - the range of experimentation could be wider (e.g. different numbers of GPUs) but this is expensive!

Overall the novelty is in making this approach work well in practice, and demonstrating it experimentally.